

# Forelimb joints contribute to locomotor performance in reindeer (*Rangifer tarandus*) by maintaining stability and storing energy

Guoyu Li, Rui Zhang, Dianlei Han, Hao Pang, Guolong Yu, Qingqiu Cao, Chen Wang, Lingxi Kong, Wang Chengjin, Wenchao Dong, Tao Li and Jianqiao Li

Key Laboratory of Bionic Engineering, Ministry of Education, Jilin University, Changchun, People's Republic of China

Corresponding author
Rui Zhang, zhangrui@jlu.edu.cn

## ABSTRACT

Reindeer (*Rangifer tarandus*) have lengthy seasonal migrations on land and their feet possess excellent locomotor characteristics that can adapt to complex terrains. In this study, the kinematics and vertical ground reaction force (GRF) of reindeer forelimb joints (interphalangeal joint *b*, metacarpophalangeal joint *c*, and wrist joint *d*) under walk, trot 1, and trot 2 were measured using a motion tracking system and Footscan pressure plates. Significant differences among different locomotor activities were observed in the joint angles, but not in changes of the joint angles ($\alpha_b$, $\alpha_c$, $\alpha_d$) during the stance phase. Peak vertical GRF increased as locomotor speed increased. Net joint moment, power, and work at the forelimb joints were calculated via inverse dynamics. The peak joint moment and net joint power related to the vertical GRF increased as locomotor speed increased. The feet absorbed and generated more energy at the joints. During different locomotor activities, the contribution of work of the forelimbs changed with both gait and speed. In the stance phase, the metacarpophalangeal joint absorbed more energy than the other two joints while trotting and thus performed better in elastic energy storage. The joint angles changed very little (~5°) from 0 to 75% of the stance phase, which reflected the stability of reindeer wrist joints. Compared to typical ungulates, reindeer toe joints are more stable and the stability and energy storage of forelimb joints contribute to locomotor performance in reindeer.

## INTRODUCTION

Large animals like ungulates and humans exhibit better locomotor efficiency than small animals like mice (*Heglund, Fedak & Cavagna, 1982*; *Taylor, Schmidtnielsen & Raab, 1970*). This is because large animals use the ground reaction force (GRF) to store mechanical energy in their elastic feet to drive the trunk forward in a locomotor cycle. The storage and generation of elastic energy in the feet is an efficient way to reduce metabolic energy cost during locomotion (*Cavagna, Saibene & Margaria, 1964*). Stretching of compliant tendons also allows limb muscles to save energy by isometric contraction under load (*Roberts, 1997*).

Thus, the muscle–tendon units of foot joints in large animals serve important functions in energy storage, stabilization, and shock absorption.

In horses, during different locomotor activities, such as walking and trotting, the long digital flexor tendons stretch and recoil from metacarpophalangeal (MCP) dorsiflexion and plantarflexion, leading to elastic energy storage and energy generation at the joints (*Biewener, 1998*). This action is like a passive spring and benefits the forelimbs during locomotion (*Bobbert et al., 2007*). The MCP joints are controlled by long tendons, superficial and deep digital flexor tendons, accessory ligaments, and muscles. These tendons are ideal structures for energy storage and generation (*Batson et al., 2003*).

The stability of foot joints is associated with animal locomotion. For a trotting horse, both the elbows and shoulders of the forelimbs have net extension moments, but there is little joint movement when the moments are maximized. These joints are relatively rigid, which allow the trunk muscles to absorb and transmit energy (*McGuigan, 2003*). The soft tissues at animal joints have important functions during locomotion. Additionally, kinematic parameters, such as joint angle, joint speed, and plantar pressure, all differ depending on locomotor speed and gait (*Abourachid, 2003*; *Kim, Rietdyk & Breur, 2008*).

Gait refers to the pattern and order of limb locomotion. Most animals use different gaits based on their terrain and locomotor speeds (*Pandy et al., 1988*; *Giovanna, Ivanenko & Lacquaniti, 2018*). Gaits are categorized by order of ground contact into walking, trotting, and galloping (*Nanua & Waldron, 1995*). Quadrupeds use different gaits depending on their locomotor speeds and select walking, trotting, or galloping during low-, moderate-, and high-speed locomotion activities, respectively. While walking and trotting, the limbs are in symmetrical gaits, and the left and right limbs are almost under constant relative phases with at least one limb in the stance phase. While galloping, the limbs are in asymmetrical gaits. As the left and right limbs change the relative phase with locomotor speed, a swing phase exists in which the four limbs are simultaneously in the air (*Alexander & Jayes, 1983*). Studies on the gaits of horses, deer, and cheetahs in terms of mechanics, energy, kinematics, and dynamics have clarified that the mechanism of gait selection corresponding to locomotor speed is related to animal balance, speed, and energy saving (*Farley & Taylor, 1991*; *Hildebrand, 1977*; *Hildebrand, 1989*; *Hoyt & Taylor, 1981*; *Minetti et al., 1999*).

Based on locomotion speed and GRF, researchers have used various methods for studying the dynamics of animal limbs in detail. *Pandy et al. (1988)* calculated the inter-articular force, joint moment, and power of goats during different locomotion activities by using GRF and limb movement, and found that the foot inertia was small and negligible relative to the trunk inertia. *Dutto et al. (2006)* and *Dutto et al. (2004)* measured the GRF, joint angle, moment, and power while trotting, and analyzed the kinematics and dynamics of the four limbs, as well as the energy storage and consumption of tendons. Moreover, the muscle stress of horse limbs while galloping was 200–400 kpa, and long tendons and extremely short pinnate muscle fibers allowed force production to be economical and enhanced the storage of tendon elastic energy (*Riemersma, Schamhardt & Lammertink, 1985*).

Load bearing and locomotion differ between the forelimbs and hindlimbs of animals. While trotting, the maximum vertical GRFs in the forelimbs and hindlimbs of German Shepherds were ~63% and ~37% of their body weight, respectively, and the impact

on the forelimbs was significant (*Rumph et al., 1994*). While walking and trotting, the maximum vertical GRFs of the forelimbs were ~1.7 and ~1.4-times those of the hindlimbs, respectively (*Witte, 2004*). The functions of horse forelimbs also differ depending on the GRFs, as the forelimbs mainly exert a braking effect and decrease the speed and kinetic energy, while the hindlimbs mainly play propulsive roles (*Dutto, 2004*; *Merkens et al., 1993*). While trotting, the maximum vertical GRFs of the forelimbs and hindlimbs are ~10-times the horizontal reaction and lateral reaction forces, while the vertical GRFs are much larger than the other component forces (*Merkens et al., 1993*). The GRFs of limbs in walking cows and gibbons are the same (*Tol et al., 2003*; *Vereecke et al., 2005a*).

Reindeer, a typical Arctic migratory animal, have a limb structure suitable for migration in complex environments (*Zhang et al., 2019*). They can adapt to various terrains, such as ice, snow, wetland, and sand (*Wareing et al., 2011*). Reindeer seasonally migrate long distances on land and some populations migrate farther than other terrestrial mammals (*Fancy et al., 2010*). Additionally, all fibers in reindeer skeletal muscles have a high oxidative capacity, which may be related to endurance activity (*Essén-Gustavsson & Rehbinder, 1985*). The sizes and structures of foot soles differ between forelimbs and hindlimbs (*Zhang et al., 2017*). The foot soles of the forelimbs are longer than the hindlimbs (87.0 ± 1.6 vs. 74.6 ± 1.0 cm). We speculate that this difference may be attributed to the different functions between reindeer forelimbs and hindlimbs during long migrations.

The toe and wrist joints of reindeer forelimbs are more stable than typical ungulates, and the stability of the wrist joint is higher. Reindeer MCP joints play the same energy storage role as in typical ungulates. Additionally, the work contribution from the forelimbs changed with gait and speed. In this study, the plantar pressure, kinematics, net joint power, and locomotor strategy of reindeer forelimbs were investigated using the vertical GRFs and limb movements during different locomotion activities. Based on previous studies, we used inverse dynamics and the static approach to explore the functions of the main forelimb joints, including energy saving and stabilization. We investigated whether the functions of interphalangeal joint *b*, MCP joint *c*, and wrist joint *d* in reindeer forelimbs were related to energy saving and stabilization during different locomotion activities. Depending on the locomotor posture and speed, reindeer locomotion was classified as walk, trot 1, or trot 2. In different locomotion activities, the temporal changes of plantar pressure and joint angles in the right forelimbs of four healthy adult male reindeer were measured. Based on inverse dynamics, we calculated the net joint moment and net joint power of the right forelimbs and analyzed the energy absorption and generation of the limbs at the joints.

## MATERIAL AND METHODS

### Samples

Fifteen healthy 8-year-old adult reindeer, including seven males and eight females, were selected from the Evenki ethnic group in Genhe City, China. The female reindeer were all excluded to rule out sex differences. Three males rushed to the fence and suffered foot injuries during training, and thus were not used in the experiment. Finally, four other easily-trained and healthy male reindeer were selected as experimental subjects. The

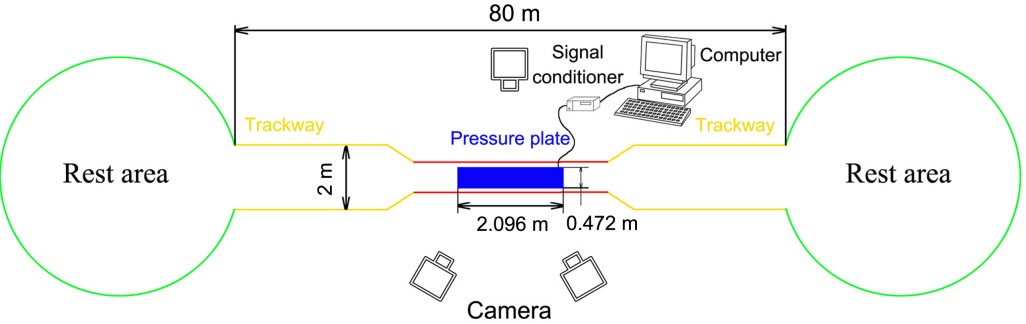

**Figure 1** **Schematic diagram of the test site.** The red line indicates the data acquisition area ($3 \times 1.5$ m$^2$), the yellow line indicates the runaway (77 m long), and the green circles indicate the rest areas. A fence (1.5 m high) was set around all areas.

animal laboratory of Jilin University has granted ethical approval (No. 3130068) to this experiment. And the field experiments were approved by the Breeding Garden of Reindeer. The masses, shoulder widths, and body lengths of the subjects were $118.75 \pm 14.93$ kg, $1.22 \pm 0.51$ m, and $1.89 \pm 0.83$ m, respectively (mean $\pm$ standard deviation). All subjects were in healthy condition and had not undergone any surgical treatment or other invasive procedures. The reindeer were kept in an outdoor fenced area ($2 \times 10^3$ m$^2$) during the day with adequate supply of food and water, which was close to a semi-wild state, and were released into a wild forest at night. Before data collection, each reindeer was trained to walk and trot on the runway, for no less than 30 min, twice a day for one month. Four right forelimbs of naturally dead reindeer were purchased and amputated from the wrist joints. The lower half of each forelimb was saved and sent for CT scans.

### Preparing the field

The field was 80 m long surrounded by 1.5-m-high fences with a 3-m-long and 1.5-m-wide data acquisition area in the middle (Fig. 1). The outside of the data acquisition area was a 77-m-long and 2-m-wide hard ground runway. The stones, weeds, and other debris were removed to ensure that the runway was even and there were places saved for reindeer to rest and eat at both ends. A pressure plate ($2,096 \times 472$ mm$^2$, 500 Hz sampling, 16384 sensors with $0.5 \times 0.7$ cm$^2$, USBII interface; Olen, Belgium) was placed on the runway and its position was adjusted to ensure that it was on the same plane as the runway. The pressure plate was connected via a signal conditioner (National Instruments, Austin TX, USA) to a computer (Dell, Xiamen, China) to record the data. One camera was placed on one side of the data acquisition area and another two cameras were placed on the other side. A high-speed camera system involving three synchronous digital cameras (Casio Exilim EX-FH25, Tokyo, Japan; 120 frame s$^{-1}$) was established. Prior to the experiment, a 36-point, three-dimensional calibration frame, located in the plane of movement over the force platform, was recorded for calibration.

During the procedures, the feeder used food or training instructions to guide the reindeer to walk and trot steadily on the runway. Adequate rest and food were provided for the animals within this period to prevent fluctuations in the test data. The locomotion of

reindeer was divided by the gaits and speeds into walk ($u = 0.44 \pm 0.08$), trot 1 ($u = 0.95 \pm 0.15$), and trot 2 ($u = 1.46 \pm 0.24$).

## Markers and joint angles

We tested the three-dimensional (3D) coordinates of the five joints (*a*, *b*, *c*, *d*, *e* (Fig. 2A)) in the right forelimbs and three joint angles ($\alpha_b$, $\alpha_c$, $\alpha_d$) by using a three-camera motion tracking system (Simi Motion 2D/3D® 7.5 software, SIMI Reality Motion Systems GmbH, Germany). The right forelimbs of the four adult reindeer underwent CT scans, and a 3D geometric model (Fig. 2C) of metacarpal, the second, third, fourth and fifth digits was established. Markers *a* (the dorsal of the hoof at the third digit), *b* (the proximal phalanx and the middle phalanx of the third digit at the joint), *c* (MCP joint), and *d* (wrist joint) were located according to the 3D limb model. The location of *e* (elbow joint) was determined based on the joint anatomy.

Regular circular reflective stickers ($R = 1.5$ cm) were used as markers which were attached to the main joints of their right forelimbs (Fig. 2A). Researchers found that the relative locations of the distal phalanx and the middle phalanx were almost on one straight line (*Doan et al., 2010*). Since the distal phalanx inside the hoof was hard to measure, the hoof and distal phalanx were taken as one part (thepart surrounded by dotted lines in Fig. 2C) and marker *a* on the hoof was considered as the joint of the middle phalanx and the distal phalanx.

We defined three joint angles (Fig. 2E), including the joint angles between the middle phalanx and the proximal phalanx ($\alpha_b$), between the proximal phalanx of the third digit and the metacarpal ($\alpha_c$), and between the metacarpal and radius ($\alpha_d$).

## Vertical GRF

The vertical GRF of each right forelimb was measured by the pressure plate. Before the measurement, the subject's weight was input into the computer and then the subject moved on the pressure plate to complete calibration. The pressure data for the right forelimb were collected and analyzed, using Footscan 7Gait 2nd generation (RSscan International, Oren, Belgium). Footscan was used to export the fore–aft coordinates of the COP during the full stance phase duration. We calculated the average path of the centre of pressure (COPy; fore–aft component) for a series of forefoot sequences within the same speed range. In terms of the kinematic data, when the displacement of the ungula cusps on $Z$-axis changed from a negative value to 0, it was identified as the touch-down moment of pressure plate data. When the displacement of the ungula cusps on $Z$-axis changed from 0 to a positive value, it was identified as the lift-off moment of pressure plate data (Fig. 2). Taking the ungula cusps as the origin, the coordinates of the COP and the joints relative to the ungula cusp were calculated, and then a global coordinates of the kinematic data and the COP was established. The relationship between GRF and time was drawn and normalized to the sample mass. Angular velocity, stance phase, vertical GRF, net joint moment, power, and work were calculated on Origin Pro 2015 (OriginLab Corporation, Northampton, MA, USA) based on the data from the joint 3D coordinates.

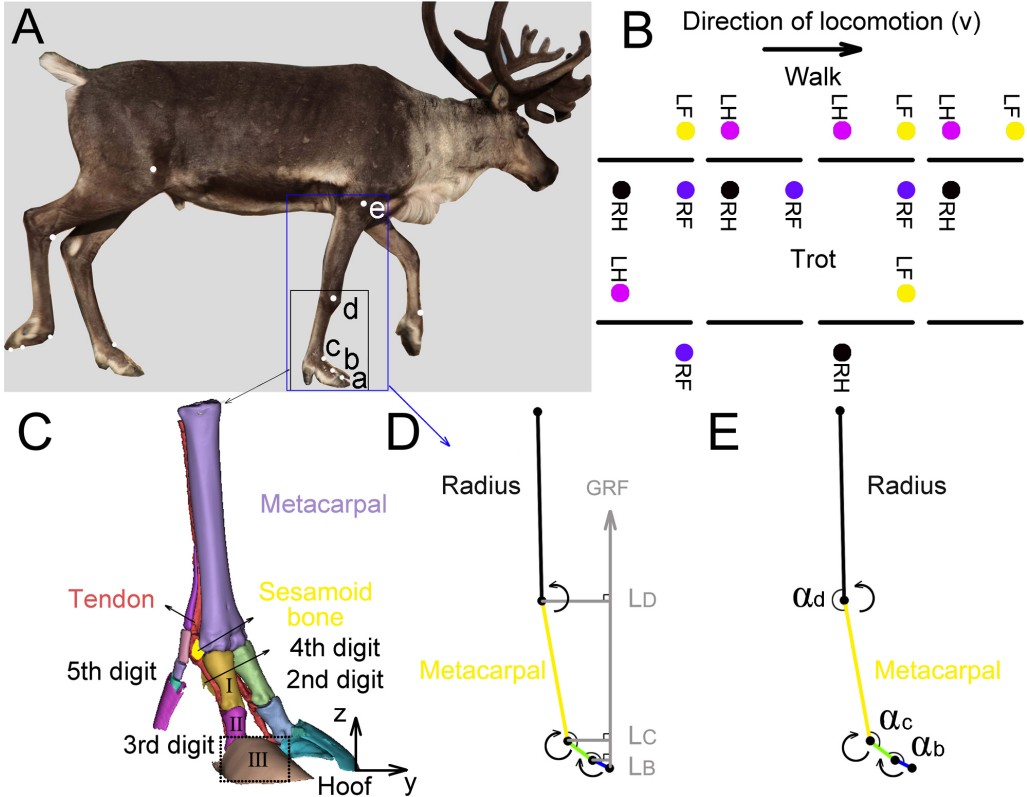

**Figure 2** **Reindeer locomotor gait, position of right forelimb joint, and schematic diagram used to calculate net moment and power.** (A) Markers on the reindeer right forelimb (*a, b, c, d, e*). (B) Model distinguishing between reindeer walking and trotting gait. RF, RH, LF and LH are right forefoot, right hindfoot, left forefoot and left hindfoot, respectively. (C) Three-dimensional model of the limb based on CT scan image reconstruction of the adult male reindeer right forelimb to determine the locations of the markers (*a, b, c, d*). (D) Reindeer limb model for calculating the net joint moment $M_m$. Net joint moment is the product of the GRF and force arm *L*. *L* is the vertical distance from the center of rotation of the joint to the GRF. The arrow indicates that the muscle produced a positive net moment direction at the joint. E. Reindeer limb model for calculating the net joint power $P_m$. The angular velocity of the joint was calculated from the derivative of the angle ($\alpha_b, \alpha_c, \alpha_d$) that changed over time. The arrow indicates the positive direction of the angular velocity. This model is the state of the reindeer right forelimb in the 25% of the stance phase. Definition: *a*. the dorsal hoof of the third digit; *b*. the joints of the proximal phalanx and middle phalanx of the third digit (interphalangeal joint); *c*. the joint of the proximal phalanx of the digit and metacarpal (metacarpophalangeal joint); *d*. wrist joint; *e*. elbow joint; I. the proximal phalanx; II. the middle phalanx; III. the distal phalanx.

## Net joint moment

The mass of the animal foot is small and the toe joints (the joint of the middle phalanx and the distal phalanx, the joint of the proximal phalanx and the middle phalanx) and wrist joint during the stance phase displaced less than other proximal joints (*Roberts, 2004*; *Vereecke & Aerts, 2008*). Therefore, we applied a static approach regardless of gravity and inertia. The net joint moment ($M_m$) was determined by vertical GRF and joint position (Fig. 2D) and was equal to the product of vertical GRF (averaged from the four reindeer at walk, trot 1, and trot 2) and *L* (vertical distance vector from the joint marker to the

GRF) (*Biewener, 1989*):

$$M_m = GRF \cdot L. \tag{1}$$

We have defined the positive direction of the forelimb joint moment of the reindeer (Fig. 2D):

For the wrist joint ($d$), net extension moment is positive (produced by extensor muscle), and net flexion moment is negative (produced by flexor muscle);

For the joints of the toes ($b$, $c$), net flexion moment is positive (produced by the plantar flexor muscle) and net extension moment is negative (produced by the plantar extensor muscle).

## Net joint power and work

To estimate the energy absorbed and generated by the interphalangeal joint $b$, the MCP joint $c$, and the wrist joint $d$, we calculated the net joint power. Joint angular velocity was calculated from the joint angle versus the time derivative by using a differential function (the central difference method). The positive direction of angular velocity is the same as that of the joint moment (Fig. 2E). The net joint power ($P_m$) of the joint equals the product of net joint moment ($M_m$) and joint angular velocity ($\omega$), where $\omega$ is averaged from the four subjects at walk, trot 1, and trot 2 (*Roberts, 2004*):

$$P_m = M_m \cdot \omega. \tag{2}$$

When the directions of the joint moment and joint angular velocity are the same, the net joint power is positive, and otherwise it is negative. Positive work and negative work represent the energy generated and absorbed by muscle–tendon units respectively (*Arnold, Lee & Biewener, 2013*).

## Gait and speed

Each reindeer completed at least five groups of tests (walk, trot 1, and trot 2) on the hard ground. We combined the research on the gaits of other animals (e.g., goats and horses (*Giovanna, Ivanenko & Lacquaniti, 2018*; *Minetti et al., 1999*)) and reindeer's locomotor postures and then sorted out the gaits and order of the footprints (Fig. 1B). The reindeer's postures of the right forelimbs at walk, trot 1, and trot 2 during the stance phase were shown in Fig. 3. The moments of touch-down, mid-stance, and lift-off are 0%, 50% and 100% of the stance phase respectively.

Walk: Symmetrical gait. At any time during the stance phase, at least two limbs are on the ground and four limbs leave the ground in sequence (e.g., the leaving sequence of left rear—left front—right rear—right front) (Fig. 2B).

Trot: Symmetrical gait. Each forelimb and its diagonal hindlimb move in the same phase, and only two limbs are in the stance phase (sometimes all four legs are in the swing phase at the same time, e.g., the leaving sequence of left rear and right front-right rear and left front) (Fig. 2B).

Speed data were normalized by Froude number ($u$), where $v$ is the average velocity, $l$ is the height of the shoulder joint from touch-down to lift-off, and $g$ is the acceleration of

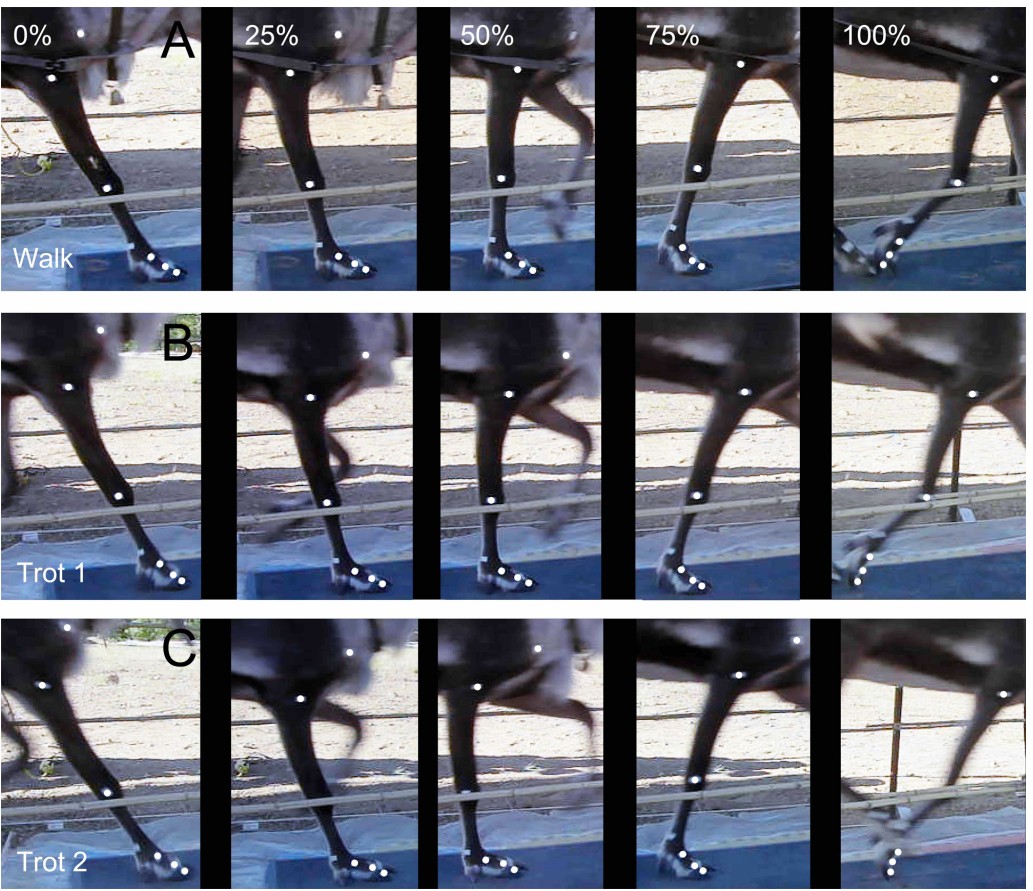

**Figure 3** **Locomotor state of the reindeer right forelimb in the stance phase.** (A) Walk. (B) Trot 1. (C) Trot 2. From left to right: moment of touch-down, 25% of the stance phase, mid-stance, 75% of the stance phase, and lift-off.

gravity:

$$u = \frac{v}{\sqrt{g \cdot l}}. \tag{3}$$

In order to examine changes with speed, relevant variables from all trotting trials were divided into two bins indicative of trot 1 (range of $u$: 0.8–1.1) and trot 2 (range of $u$: 1.1–1.7). The speeds of reindeer at walk, trot 1, and trot 2, after nondimensionalization, are 0.44 ± 0.08, 0.95 ± 0.15, and 1.46 ± 0.24 (mean ± S.D.).

## Statistical analysis

Vertical GRF, net joint moment and net joint work were normalized to stance duration and to reindeer's mass. During walk, trot 1, and trot 2 in the stance phase, the kinematics and the plantar pressure data of four reindeer were combined to analyze the temporal variation of joint angles and vertical GRFs during the stance phase. The means and the variances of the vertical distance vector from the joint marker to the GRF ($L$) and joint angular velocity ($\omega$) were obtained from four samples at walk, trot 1, and trot 2, respectively. The joint

angular velocity was the derivative of the joint angle with respect to time. Meanwhile, the means of the vertical GRF from four reindeer were obtained. Based on the above data, the net joint moment and the net joint power were calculated by using the formula. The work done at the joint was the integral of the net joint power with respect to time. Statistical analyses were conducted to examine the differences in different gaits and speeds, three key indicators (joint angles at touch-down, mid-stance and lift-off) between slow walking and trotting gaits/trot 1 and trot 2 speeds by using Origin Pro 2015 (OriginLab Corporation, Northampton, MA, USA). We used one-way ANOVA statistical technique to analyze the effect of locomotor gait and speed on each joint angle indicator. $F$-test was conducted to examine whether these two variations are significantly different. Statistical significance level was considered as $P < 0.05$.

# RESULTS

## Joint angles

During walk, trot 1, and trot 2 in the stance phase, the reindeer interphalangeal joint angle $\alpha_b$ (Figs. 4A–4C), MCP joint angle $\alpha_c$ (Figs. 4D–4F), and wrist joint angle $\alpha_d$ (Figs. 4G–4I) showed similar patterns and ranges. Therefore, the data of joint angles of the four reindeer were combined to analyze the temporal variation of joint angles during the stance phase. The stick figure of the forelimbs during different locomotor stance phases was shown in Figs. 4J–4L. At the moment of touch-down, the limb joints ($b, c, d$) first moved toward the ground and then left the ground after touching the lowest point. This pattern of motion may be related to energy saving.

The integrated data correspond to the means and variances of $\alpha_b$, $\alpha_c$ and $\alpha_d$ (Figs. 5A–5C) at walk, trot 1, and trot 2 in the stance phases. The joint angles ($\alpha_b$, $\alpha_c$, $\alpha_d$) displayed similar patterns during different locomotor stance phases. The $\alpha_b$ increased (joint plantarflexion) during the early stance phase (about 0–30%), decreased (joint dorsiflexion) in the mid-stance phase (about 30%–80%), and rose (joint plantarflexion) in the late stance phase (about 80%–100%). The interphalangeal joint $b$ plantarflexed in the late stance phase, and the hoof gradually lifted off the ground with the tip still in contact with the ground.

During different activities, the maximum and minimum values of $\alpha_b$ and $\alpha_c$ during the stance phase and the corresponding time points differed. Also the joint range of motion (ROM) was larger at the trotting gaits than at the walking gait. The ROMs of $\alpha_d$ among the three joint angles were the greatest, around 29°, 30° and 35° during walk, trot 1 and trot 2, respectively. The ROMs of $\alpha_b$ were the smallest, around 26°, 27° and 33° during walk, trot 1 and trot 2, respectively. Therefore, the relationships of joint angles with gaits and speeds showed that the joints can adapt to different gaits and locomotor speeds.

## Vertical GRF

The forelimbs have different vertical GRFs at different time points during the stance phase. According to the time corresponding to the peak vertical GRF, the forelimb locomotion can be divided into a braking phase and a propulsive phase (*Vereecke et al., 2005b*). The

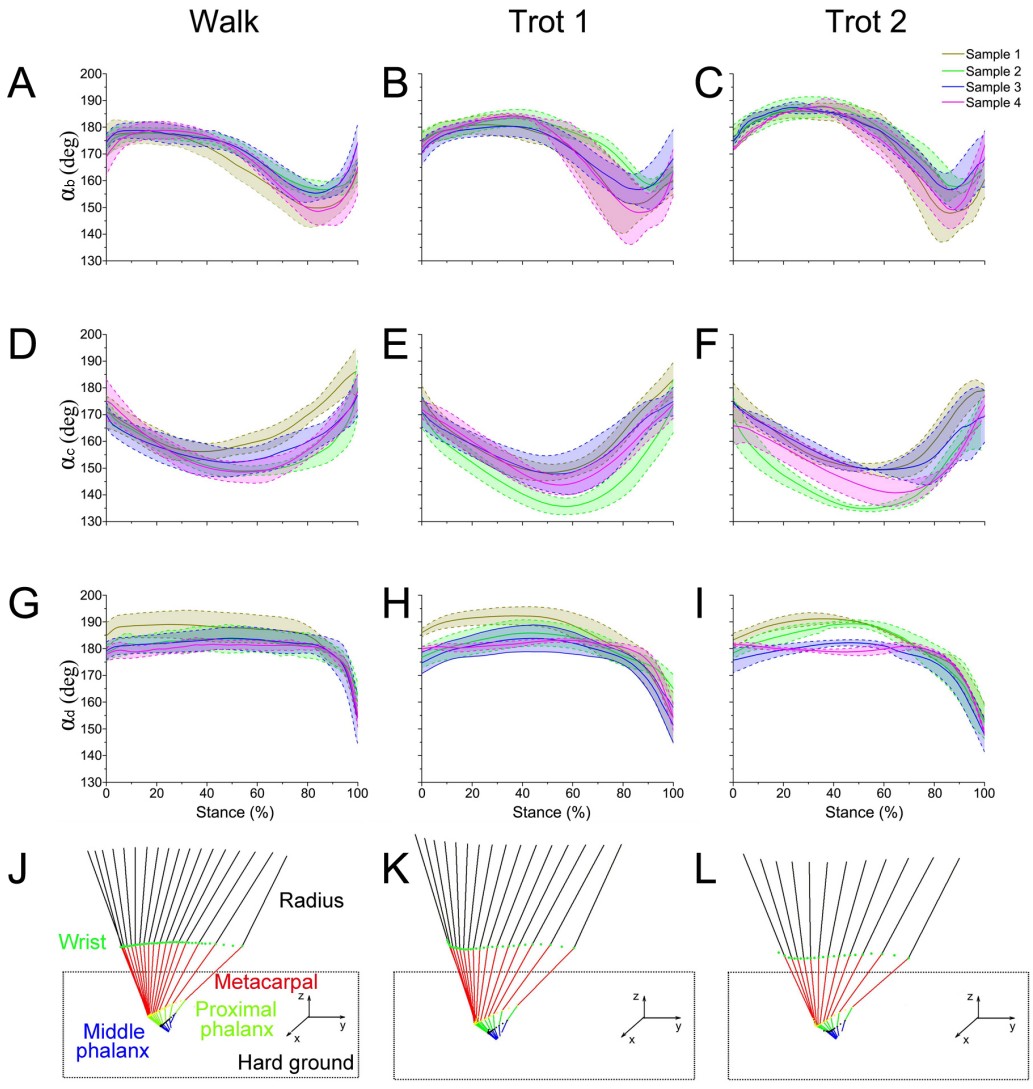

**Figure 4** **The means and standard deviations of the joint angles ($\alpha_b$, $\alpha_c$, $\alpha_d$) from the four samples.**
(A, B, C) Curve of joint angle $\alpha_b$ during the stance phase at walk, trot 1, and trot 2, respectively. (D, E, F) Curve of joint angle $\alpha_c$ during the stance phase at walk, trot 1, and trot 2, respectively. (G, H, I) Curve of joint angle $\alpha_d$ during the stance phase at walk, trot 1, and trot 2, respectively. (J, K, L) Motion of the right forelimbs during the stance phase at walk, trot 1, and trot 2, respectively. Yellow, green, blue, and pink lines represent reindeer samples 1, 2, 3, and 4, respectively.

vertical GRF increased with time during the braking phase, and decreased with time during the propulsive phase (Fig. 6).

The peak vertical GRFs (normalized to body mass) during walk, trot 1, and trot 2 were 8.95, 11.33, and 12.80 times the body mass, respectively, and the corresponding peak time was 57.03%, 50.45%, and 47.78% of the stance phase, respectively. The gaits and locomotor speeds of reindeer affect the vertical GRF. As for different gaits, the peak vertical GRF at trot was larger than that at walk. At the same gait, the peak vertical GRF at trot 2 was larger than that at trot 1.

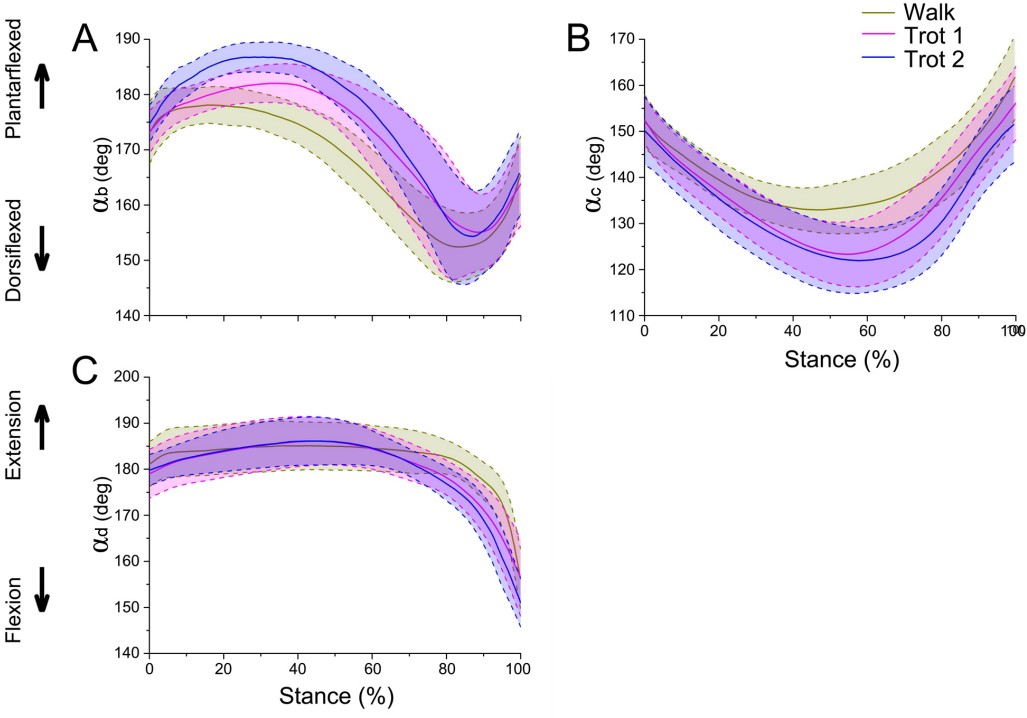

**Figure 5** **Means and standard deviations of the three joint angles in walk, trot 1, and trot 2 locomotion on hard ground during the stance phases.** (A) Curve of joint angle $\alpha_b$ over time. (B) Curve of joint angle $\alpha_c$ over time. (C) Curve of joint angle $\alpha_d$ over time. The yellow, pink, and blue lines represent walk, trot 1, and trot 2, respectively.

## Net joint moment

In different activities, the forelimb joints $b$ and $c$ of reindeer in the stance phase (about 0 to 100%) produced positive net flexion moment by the plantar flexor (Fig. 7). Joint $d$ in the early stance phase (about 0 to 75%) and late stance phase (about 75 to 100%) generated the negative net flexion moment and positive net extension moment respectively by the flexor and extensor muscles (Fig. 8). Reindeer and horses have similar net joint moment curves for joints $c$ and $d$ when trotting on hard ground (*Dutto et al., 2006*).

In different activities, reindeer have different peak net joint moments at the forelimb joints. Joints $b$, $c$, and $d$ reached the peak net joint moments at about 45%, 50% and 30% of the stance phase, respectively. The peak net joint moments at walk, trot 1, and trot 2 were 0.28, 0.37, and 0.42 Nm kg$^{-1}$ at joint $b$, 0.55, 0.79, and 0.93 Nm kg$^{-1}$ at joint $c$, and −0.95, −1.35, and −1.78 Nm kg$^{-1}$, at joint $d$, respectively. The vertical GRF of reindeer forelimbs increased with the rising locomotor speed and, accordingly, the peak joint moment also rose. Since the vertical distance vector of vertical GRF from joint $d$ was the largest, joint $d$ had the greatest peak joint moment, followed successively by joint $c$ and joint $b$.

## Net joint power and work

The net joint moment reflects the activity (extension and flexion) of muscles (extensors and flexors) , but not the changes of energy in the muscle–tendon units at the joints. Net

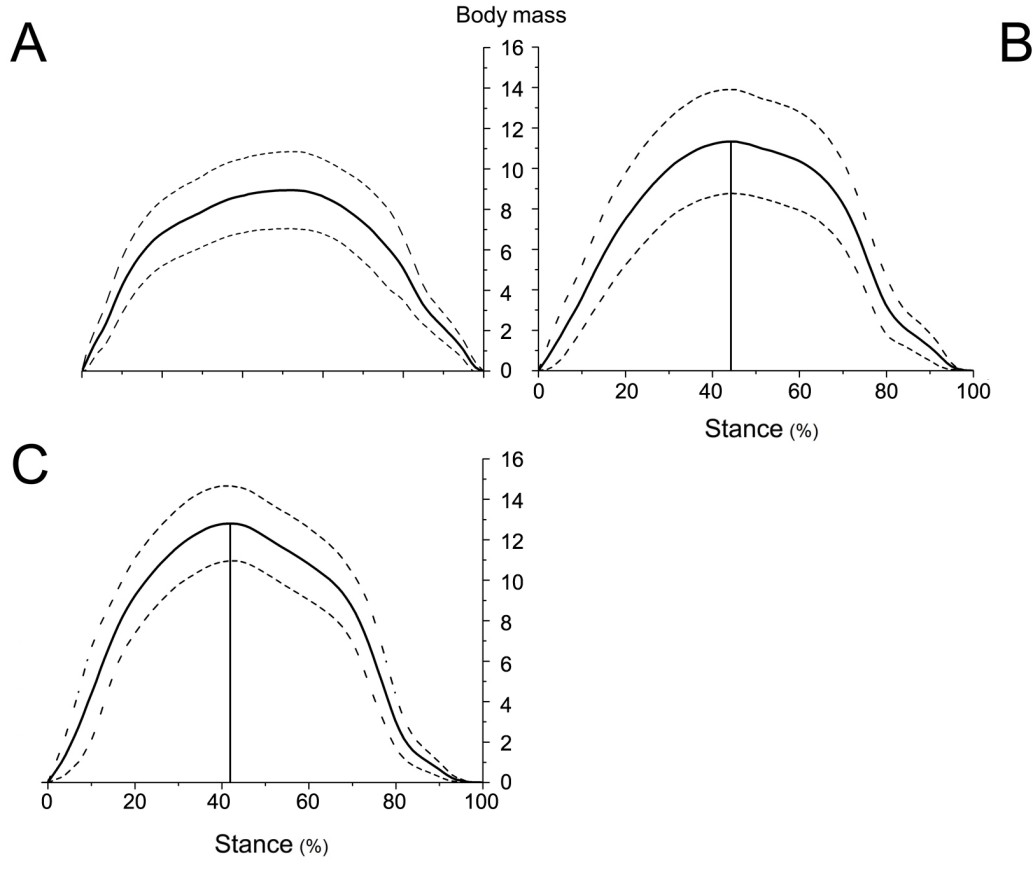

**Figure 6** Means and standard deviations of the vertical GRFs (normalized to body mass) of the right forelimbs on hard ground in walk (A), trot 1 (B), and trot 2 (C) during the stance phase.

joint power and work, which are directly related to the energy absorption and generation at the limb joints (*Harrison et al., 2010*), at the forelimb joints were shown in Figs. 7 and 8. As mentioned above, the net joint moment and angular velocity at the forelimb joints increased with the rise of locomotor speed. Therefore, the net joint power at the joints increased accordingly. The net joint power ranges at joint $c$ were the largest and were −0.37 to 0.06, −0.19 to 0.21, and −4.37 to 2.46 W kg$^{-1}$ at walk, trot 1, and trot 2, respectively. As the locomotor speed was accelerated, the net joint power range was enlarged and thus the feet need to absorb and generate more energy at the joints.

In different activities, reindeer had similar work patterns at the same joint. From about 0 to 55% of the stance phase, the dorsiflexion of joint $c$ produced a net flexion moment and the foot absorbed energy at the joint. From about 55% to 100% of the stance phase, joint $c$ plantarflexed and the plantar flexor and extensor muscles generated and absorbed energy, respectively (details of energy absorption and generation at each joint are shown in Table 1). The energy changes at the limb joints are related to joint functions, such as energy storage and stabilization (*Biewener, Thomason & Lanyon, 1988*; *Lee, 2011*).

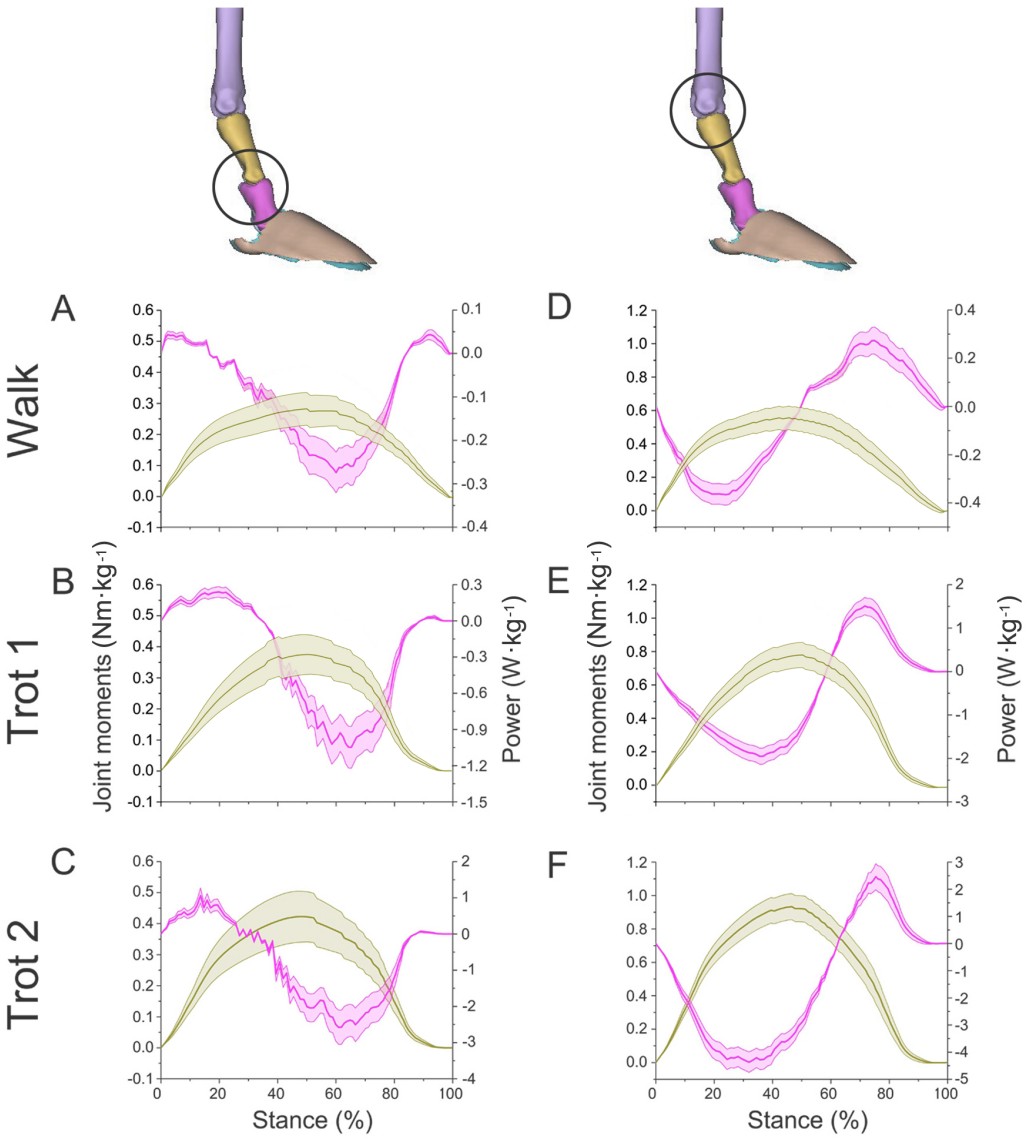

**Figure 7** **Means and standard deviations of net joint moments (yellow solid line, mean; yellow shaded area, standard deviation) and net joint power (pink solid line, mean; pink shaded area, standard deviation) of the right forelimb joints.** (A, B, C) The net joint moment and net joint power of interphalangeal joint *b* in walk, trot 1, and trot 2. (D, E, F) The net joint moment and net joint power of MCP joint *c* in walk, trot 1, and trot 2. Interphalangeal joint *b* and MCP joint *c* are indicated by the left and right circles, respectively.

## DISCUSSION

We investigated whether the functions of interphalangeal joint *b*, MCP joint *c*, and wrist joint *d* in the forelimbs correlated with energy saving and stability. Depending on the locomotor postures and speeds, reindeer locomotor activities were divided into walk, trot 1, or trot 2. In different locomotion activities, we measured the temporal changes of plantar pressure and joint angles in the right forelimbs of four healthy adult male reindeer. Based

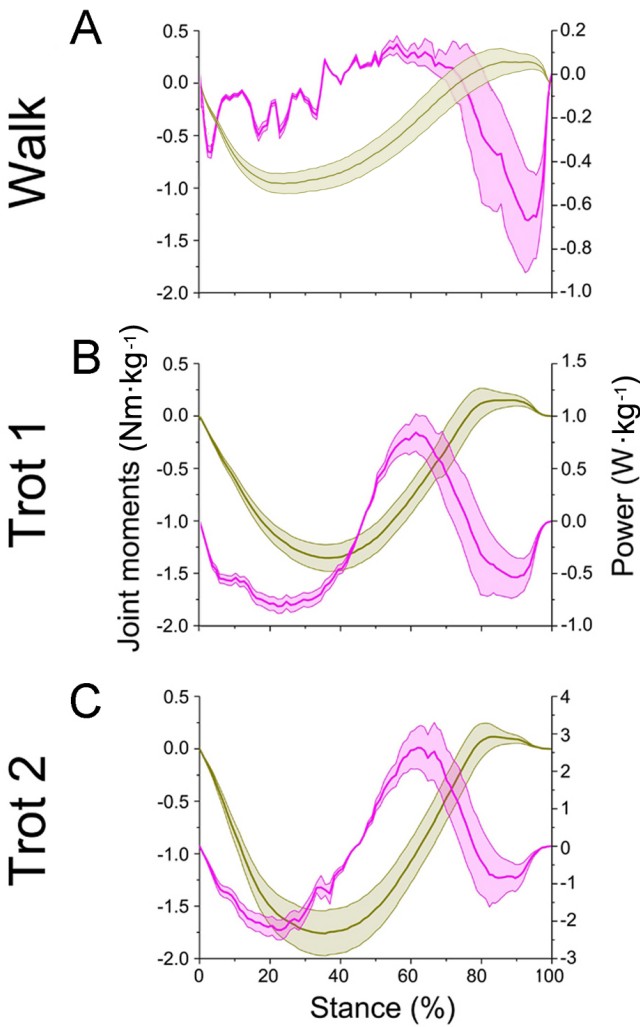

**Figure 8 Means and standard deviations of net joint moment (yellow solid line, mean; yellow shaded area, standard deviation) and net joint power (pink solid line, mean; pink shaded area, standard deviation) at wrist joint *d* of the right forelimb.** (A, B, C) The net joint moment and net joint power of wrist joint *d* in walk, trot 1, and trot 2, respectively.

on inverse dynamics, we calculated the net joint moment and net joint power of the right forelimbs, and the energy absorption and generation by the limb joints.

Locomotor strategies change depending on the speed and gait adopted by the animal. For example, goat limbs adjust the work of muscles and tendons to adapt to a walk or trot locomotion on slope-variable surfaces (*Mcguigan et al., 2009*). Horses rely on the minimization of metabolic costs and change gaits based on a range of different locomotion speeds, including the most energy-efficient trot gait (*Hoyt & Taylor, 1981*). In our study, during different gaits (walk or trot) and speeds (trot 1 or trot 2), significant differences were detected in the reindeer joint angles at the moments of touch-down, mid-stance, and lift-off (Fig. 9), which may be associated with reindeer locomotor strategies. Although reindeer changed locomotor strategies during different gaits and speeds, we still found

**Table 1  Positive, negative, and net power produced by reindeer forelimbs joints during different locomotion activities.**

| Joint | Locomotion | Positive power (J kg$^{-1}$) | Negative power (J kg$^{-1}$) | Net power (J kg$^{-1}$) |
|---|---|---|---|---|
|  | Walk | $5.84 \times 10^{-3}$ | $-6.09 \times 10^{-2}$ | $-5.91 \times 10^{-2}$ |
| b | Trot 1 | $1.50 \times 10^{-2}$ | $-10.63 \times 10^{-2}$ | $-8.97 \times 10^{-2}$ |
|  | Trot 2 | $2.95 \times 10^{-2}$ | $-18.79 \times 10^{-2}$ | $-14.50 \times 10^{-2}$ |
|  | Walk | $4.64 \times 10^{-2}$ | $-6.42 \times 10^{-2}$ | $-2.74 \times 10^{-2}$ |
| c | Trot 1 | $9.58 \times 10^{-2}$ | $-19.71 \times 10^{-2}$ | $-15.00 \times 10^{-2}$ |
|  | Trot 2 | $9.52 \times 10^{-2}$ | $-33.03 \times 10^{-2}$ | $-29.50 \times 10^{-2}$ |
|  | Walk | $1.39 \times 10^{-2}$ | $-10.69 \times 10^{-2}$ | $-8.77 \times 10^{-2}$ |
| d | Trot 1 | $5.00 \times 10^{-2}$ | $-9.98 \times 10^{-2}$ | $-6.44 \times 10^{-2}$ |
|  | Trot 2 | $12.02 \times 10^{-2}$ | $-15.09 \times 10^{-2}$ | $-4.80 \times 10^{-2}$ |

some similarities, such as the elastic energy saving function of joint $b$ and the effect of joint $d$.

## Contribution of work change with gaits and speeds

Most animals use the inverted pendulum model in their walking gaits and restore mechanical energy via the periodic conversion of kinetic and potential energies (*Donelan, Kram & Kuo, 2002*; *Full & Koditschek, 1999*). While trotting, the spring-mass system and inverted pendulum model are used, wherein the limbs act as springs that store and generate energy, which is characterized by a significant reduction in the difference between the potential and kinetic energies during the stance phase (*Geyer, Seyfarth & Blickhan, 2006*; *Farley, Glasheen & Mcmahon, 1994*). In our study, significant differences were detected in joint angles $\alpha_b$ and $\alpha_c$ between walking and trotting gaits, and in $\alpha_b$ and $\alpha_d$ between trot 1 and trot 2 (Fig. 9). In the trotting gait, the MCP joint absorbed more energy than the other two joints (Table 1), but in the walking gait, the MCP joint absorbed less energy than the wrist joint. This may be attributed to the preference of animals over the inverted pendulum gait at low speeds and over the mass spring inverted gait at high speeds, which both enhanced locomotor performance and energy saving (*Cavagna, Thys & Zamboni, 1976*; *Muir, Gosline & Steeves, 1996*). The different motion patterns among different gaits and speeds in reindeer forelimbs may be caused by the more efficient energy mechanism.

Reindeer have enhanced MCP joints. While walking, the proximal phalanx pivots about joint $b$ (stance phase of 0–10%) with slight downward and upward movements (Fig. 5A). However, while trotting, the distal phalanx moves downward (joint $b$ plantarflexion) for a prolonged period of time (stance phase of 0–20%) (Fig. 5A). Owing to the stretching and recoiling of plantar flexor tendons, plantarflexion and dorsiflexion of interphalangeal joint $b$ are typical in loading and rebounding patterns. This indicates that the elastic elements at the toe joints offset the GRF and the trotting gait can reduce pressure, as well as protect the soft tissues of the toes by prolonging the foot-to-ground contact time ratio.

As previously reported in horses, as locomotor speed increased, the positive and negative work done by MCP joint $c$ increased significantly, and elastic energy storage and generation also increased (*Dutto et al., 2006*). This finding is consistent with our results. When the

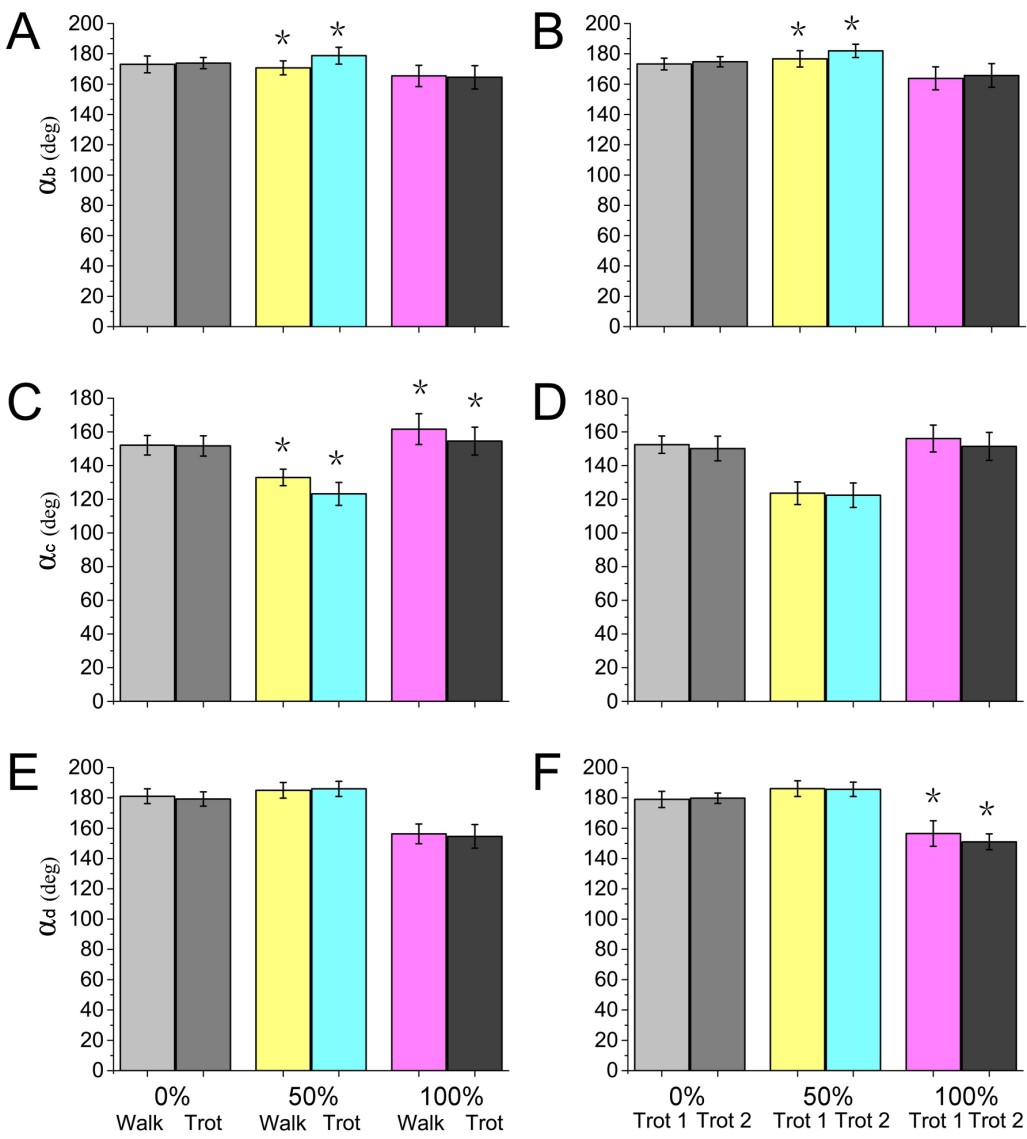

**Figure 9** **Numerical statistical analysis of the effect of locomotor gait and speed to the forelimb joint angles ($\alpha_b$, $\alpha_c$, $\alpha_d$) of the reindeer at the touch-down (0%), mid-stance (50%), and lift-off (100%).** (A, C, E) Analysis of the significant effects of walking and trotting gait on joint angles $\alpha_b$, $\alpha_c$, and $\alpha_d$. (B, D, F) Analysis of the significant effects of the locomotor speed of trotting gaits on joint angle $\alpha_b$, $\alpha_c$, and $\alpha_d$. *$P < 0.05$.

ROM of MCP joints and net flexion moment increased as locomotor speed increased, the foot work of reindeer at the joint also increased. The difference in the ROM of wrist joint $d$ was small among different activities (~5°), but as locomotor speed increased, the vertical GRF and angular velocity also increased, thereby increasing the net joint moment and net joint power. Compared to the trotting gait, the power of the wrist joint was smaller, but fluctuated more severely in the walking gait (Fig. 8A). Thus, slow locomotor activities may require a higher level of neural control (*Schaller et al., 2011*).

## Comparison of reindeer forelimb joints to typical ungulates

Ungulate locomotion has evolved in vastly different patterns depending on the specific habitat of a given species (*Hildebrand, 1976*). Compared to horse forelimbs, reindeer toe joints are stable, as the ROMs between the interphalangeal and MCP joints are smaller. The ROMs of interphalangeal joint *b* in reindeer and horses are ∼30° and ∼40°, respectively, and the ROMs of MCP joint *c* in reindeer and horses are ∼31° and ∼40°, respectively (*Dutto et al., 2006*). These differences indicate that reindeer forelimbs have stable toe joints. Measurements obtained using a linked segment model and spring coefficients from a spring model demonstrated that the stiffness of goat limbs was twice that of dog limbs during different activities, suggesting that goats have adapted to a rougher and steeper terrain (*Lee et al., 2014*). Wrist joint *d* flexion produced a net flexion moment, which generated propulsion in the middle stance phase (45–75%) (*Ishida, 1984*). In the last 20% of the stance phase, the long digital flexor tendons at interphalangeal joint *b* and MCP joint *c* recoiled, and joint plantarflexion produced a net flexion moment; however, the net flexion moment and propulsion were small (Fig. 7). Wrist joint *d* changed between 0 and 75% of the stance phase (∼5°). While trotting, the change trend of wrist joint *d* in horses was similar to reindeer. The wrist joint angle of horses was maintained at 180–190° within 0–60% of the stance phase, then gradually decreased (*Dutto et al., 2006*). Wrist joint *d* in reindeer was maintained and stable over time. Reportedly, horse knees during the stance phase produce a net flexion moment and the flexor muscles assist foot movement, wherein the extensor muscles stabilize the joints (*Schaller et al., 2011*). Similarly, reindeer wrist joint *d* displayed this stabilizing ability during the early and middle stance phases (0–75%). The small change in the joint angle (∼5°) indicated that the wrist joint plays a role in stabilizing foot locomotion. Thus, the low flexibility and high stability of the forelimb joints may be beneficial during long distance migrations.

## MCP joints as energy storage devices

The MCP joints of most animals elastically store and generate energy because they are mainly composed of small muscles, short pinnate muscle fibers, and long tendons (*McGuigan, 2003*; *Payne et al., 2005*; *Wickler, 2005*). Ligaments have a protective effect on joints (*Schaller et al., 2009*), and tendons also provide an energy advantage in high-speed locomotion (*Schaller et al., 2011*).

Reportedly, the relatively short muscle fibers and long tendons in turkey hindlimbs act like springs (*Ker, Alexander & Bennett, 1988*), as the short muscle fibers contribute to more economical muscular energy and stretching of the long tendons allows muscle fibers to generate energy with little change in length, thus decreasing metabolic costs (*Rall, 1985*; *Biewener & Roberts, 2000*). The MCP joints in reindeer feet also absorb and generate energy during different locomotor activities (Table 1), which are manifested by an elastic system for energy storage and generation (Fig. 7). The distal joints in horse forelimbs recover 40% of the energy during the stance phase (*Biewener, Thomason & Lanyon, 1988*). Furthermore, 70–80% of the plantar flexors stretched at the metatarsal joints during the stance phase. The Achilles tendons, long plantarflexion tendons, and plantar connective tissues of the feet absorb energy and convert it into elastic potential (*Vereecke et al., 2005b*). Reindeer

MCP joints at the same position in the forelimbs also performed well in energy storage during different locomotor activities and absorbed $6.42 \times 10^{-2}$, $19.71 \times 10^{-2}$, and $33.03 \times 10^{-2}$ J kg$^{-1}$ of energy (negative power) in walk, trot 1, and trot 2, respectively.

## CONCLUSIONS

The forelimb joint angles of reindeer ($\alpha_b$, $\alpha_c$, $\alpha_d$) changed in similar patterns during different locomotor stance phases. The peak vertical GRF increased as locomotor speed increased. The peak vertical GRFs (normalized to body mass) in walk, trot 1, and trot 2 were 8.95, 11.33, and 12.80-times the body mass, respectively.

During different locomotor activities, the joint angles significantly differed at the touch-down, mid-stance, and lift-off moments. In the trotting gaits, the MCP joint absorbed more energy than the other two joints, but in the walking gaits, it absorbed less energy than the wrist joint. Across different gaits and locomotor speeds, the forelimbs adopted different locomotor strategies to improve locomotor performance and save energy.

As reindeer speed increased, the peak joint moment and net joint power both increased. The feet absorbed and generated more energy at the joints. The feet first absorbed energy, then generated energy at the MCP joint during the stance phase, thus performing well in elastic energy storage. In the middle stance phase (45–75%), the feet exerted a propulsive effect during the flexion of the wrist joint. In the early and middle stance phases (0–75%), the joint angle changed very little ($\sim5°$) and the wrist joint stabilized the feet. Clearly, during long-distance migration, forelimbs play stability maintenance and energy storage roles.

The kinematics of hindlimb and the coordination of hindlimb and forelimb of reindeer would be analyzed to study the effect of reindeer foot joint on movement in the future study.

### Funding

This work was supported by the National Natural Science Foundation of China (No. 51675221), the Science and Technology Development Planning Project of Jilin Province of China (No. 20180101077JC) and the Science and Technology Research Project in the 13th Five-Year Period of Education Department of Jilin Province (No. JJKH20190134KJ). The funders had no role in study design, data collection and analysis, decision to publish, or preparation of the manuscript.

### Grant Disclosures

The following grant information was disclosed by the authors:
National Natural Science Foundation of China: 51675221.
Science and Technology Development Planning Project of Jilin Province of China: 20180101077JC.
Science and Technology Research Project in the 13th Five-Year Period of Education Department of Jilin Province: JJKH20190134KJ.
## Competing Interests

The authors declare there are no competing interests.

## Author Contributions

- Guoyu Li and Rui Zhang conceived and designed the experiments, prepared figures and/or tables, and approved the final draft.
- Dianlei Han and Hao Pang performed the experiments, prepared figures and/or tables, and approved the final draft.
- Guolong Yu, Qingqiu Cao, Chen Wang, Lingxi Kong, Wang Chengjin, Wenchao Dong and Tao Li analyzed the data, authored or reviewed drafts of the paper, and approved the final draft.
- Jianqiao Li performed the experiments, authored or reviewed drafts of the paper, and approved the final draft.

## Animal Ethics

The following information was supplied relating to ethical approvals (i.e., approving body and any reference numbers):

The ethical examination of the animal laboratory of Jilin University passed with the ethical approval (NO. 3130068).

## Field Study Permissions

The following information was supplied relating to field study approvals (i.e., approving body and any reference numbers):

The Breeding Garden of Reindeer approved field experiments.

## Data Availability

Raw data are available in the Supplemental Files.

## Supplemental Information

Supplemental information for this article can be found online at http://dx.doi.org/10.7717/peerj.10278#supplemental-information.

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
