# Peer review of "Forelimb joints contribute to locomotor performance in reindeer (Rangifer tarandus) by maintaining stability and storing energy"

_PeerJ, doi:10.7717/peerj.10278_

## Round 0.1 · original submission · Major Revisions

Overall, the 2 reviewers see value in the study but have numerous constructive critiques that need consideration in revision. Please include a point-by-point Rebuttal in maximal detail. Thank you.

Reviewer 1 ·

Basic reporting

In general, the writing was clear. There are some word choices that should be changed. For examples a careful proof reading will find many of these:
Abstract Line 18: “…vertical GRF were increased as the locomotor…” The word increased is more important than enhanced.
Introduction, Line 54: use ‘terrain’ instead of “landforms”
Line 116: “…tested reindeer were in healthy condition and…”
Line 118: “…outdoor fenced area (2 x 10…”
Discussion, Line 312: why use the word “permanently” here?
Discussion, Line 320: “work” not “works”

The review of literature in the introduction accounts for previous work in the field. The paper is generally well referenced, however there are papers that are cited which contain more information relevant to the study than indicated. For example, the paper by Dutto et al (2004) has joint angle, moment, and power during trotting in horses and it is not in the references. The focus on comparison with other ungulates is relevant, particularly with regard to the morphology of the forelimb and associated function. Comparing morphology and mechanics of the reindeer with other, similar quadrupeds would provide a good analysis. That is only done peripherally as part of the overall paper.

Dutto, D. J., Hoyt, D. F., Cogger, E. A., & Wickler, S. J. (2004). Ground reaction forces in horses trotting up an incline and on the level over a range of speeds. Journal of Experimental Biology, 207(20), 3507-3514.

The paper follows the format guidelines and standards of PeerJ.

Figures are generally of high quality. Here some issues:
Figure 2. It was unclear where the dotted line in Fig 2C is located to indicate the distal foot. Is it the partial square? It did not come through very clear in the copy I received.
Figures 4, 5, 6,7,8: It is helpful to have all plots within a figure to have the same scale. This allows for an easier visual comparison of each joint/condition.
Figure 7 and 9. The caption is cut off so that some of the text is missing.

Finally, there is raw data supplied. However, I was not able to open all of the data files, as I got an error when attempting to do so.

Experimental design

Determining the forelimb kinetics of a cursorial, quadrupedal, ungulate is relatively novel. It appears that the research questions being addressed are:
1. “Is the function of reindeer limb joints different from other ungulates?”
2. “Do reindeer adopt different strategies across different gaits and speeds?”

These are good questions. However, neither one is answered well in the context of the paper. Some of the confusion might be with the wording of the question. For example, if the second question was “Does the contribution of work of the forelimb change with gait or speed?”, it would provide more clarity and focus on what was being discussed. Additionally, the comparison with other ungulates is relevant due to the type of terrain over which these animals must traverse. Providing more in-depth comparison with previously published work would help to illustrate the function of the limb in the animal pertinent to the environment.

The methods, in general, are appropriate for an investigation of this type. Creating a space to safely test animals such as these can be challenging, especially given the tricky nature of capturing both video and ground reaction force. The set-up is sound. With regards to the video and reconstruction of three-dimensional coordinates, there is no information given on the calibration of the capture space. Also, since just three cameras were used, obtaining accurate reconstruction of the marker trajectories can be tricky. Did the investigators validate their set-up? Were the cameras synchronized? How was the video synchronized with the pressure plate data?

It indicates that “each reindeer completed at least five groups of tests”. Does this mean that the results provided are an average of 5 trials at each gait condition for an animal? How were data combined if so? Or was just one trial used for each animal?

With regard to pressure plate, how was the center of pressure validated? How was this combined with the kinematic data? How was calibration of the entire set-up (force and kinematics) done to insure accuracy? Small discrepancies in COP location relative to marker locations can have a significant effect on joint moment calculations.

For both the kinematic and kinetic data, did any data processing/reduction occur? Were the data smoothed or resampled in any way? There is a fair amount of noise in the power data, but not in the moment or presented angle data.

How were the speed bins chosen for trot 1 and trot 2?

The sample size is very small, which is not uncommon with these types of data collection. In order to confidently indicate that differences were present, statistical analysis is required. Further, consistent indication of variability (such as standard deviation) is relevant. Using the questions posed in the introduction, perform specific statistical assessments on your data for comparison. This will strengthen any conclusions you have to make.

Validity of the findings

The observations of the study are relatively novel. Similar results have been observed from trotting in horses, dogs, goats, and other animals. However, the apparent structure of the distal limb (foot) lends import to the observations. A more detailed comparison with previous literature on quadrupeds would strengthen the findings. In terms of whether energy savings are occurring, understanding function of the proximal joints, and the general metabolic cost of transport will help to provide relevance. One other consideration is that previous work has found that the forelimb acts more for support and less for propulsion during locomotion (Dutto, et al 2006, Payne et al, 2005). I suspect that this is similar for the reindeer. The relative energy absorption of the MCP is helping to control the vertical motion. Combining these results with what is occurring mechanistically in the hindlimb, will give a better overall picture of the changes associated with gait and how similar or dissimilar these animals are from other ungulates.

Reviewer 2 ·

Basic reporting

The grammar and writing are acceptable. There are a few instances where corrections are advisable and these are noted in my comments. The authors have met criteria for references, background, professional article structure, figures and tables. They have also provided the raw data. The results are self-contained and relevant to the introduction in the article. I would only ask that they make their hypotheses or expectations explicit in the introduction (though they somewhat implicit; see General Comments).

Experimental design

This is original primary research that is within the aims and scope of the journal. The research questions are somewhat well defined (though I would like to see explicit hypotheses/expectations; see General Comments), relevant and meaningful. The authors state how this study fills a gap in the literature. The authors provided a rigorous investigation of reindeer forelimb mechanics using inverse dynamics and appear to have met institutional ethical requirements. The methods have been described sufficiently to allow for replication.

Validity of the findings

The findings in this study are valid, robust, and are sound. The authors used statistics but did not indicate what tests were used (see General Comments). The authors conclusions are well stated, though not linked to well defined hypotheses/expectations setup in the introduction. There are a few instances of speculation that need to be identified as the authors novel hypothesis or require elaboration (see General Comments).

Additional comments

Overall, this is a well done study of forelimb mechanics in reindeer. However, I do have some questions and suggestions:

I think the title needs to be more specific and relay the what the authors did in the study or what their findings were. The current title ‘Forelimb joints contributing to locomotor performance in reindeer’ is a statement that could be made without the contents of this manuscript and should be ‘Forelimb joints contribute to locomotor performance in reindeer’. In what way do they contribute?

The abstract would benefit from a conclusion statement or discussion point. This is just a summary of methods and results.

The introduction is comprehensive, but it took a long time to get to your question. I think a statement of the question/problem early on would help frame the introduction and background necessary to understand what we know and what we don’t know.

Did you collect data from the female reindeer and then exclude them because they showed differences compared to males or did you not collect data on them? Are there expectations that female reindeer forelimb mechanics will differ from males other than body size effects?

Is there a reason why you didn’t calculate joint work for the shoulder to characterize joint work/power for the entire forelimb?

In Fig 9, the authors present statistical results, but don’t indicate what statistical test was used. Please put this in methods.

Overall, I think this paper would benefit from some predictions or hypotheses. The introduction is comprehensive and following a discussion of previous findings in horses and other ungulates, the authors could setup a few expectations that they could use as a comparison in the results or discussion section. This could just be a couple of paragraphs in total. As is, the manuscript is fine and provides a sufficient description of their findings. However, I think the overarching conclusion that reindeer use different strategies, presumably to moderate energy costs at different speeds (which should be the null hypothesis) needs to be expanded on a little bit. The headings the authors used in the discussion (e.g., ‘Wrist joints as a stabilizer and pusher’) would be perfectly suitable as predictions that could be setup in the introduction.

Figure 2B is a little challenging to interpret. It might be easier to follow if you labeled the circles in the figure so you don’t have to keep referencing the figure legend. One option is to put labels like FL (front left) in the circle.

Figure 8. I assume Walk is (A), Trot 1 is (B) and Trot 2 is (C), but these should be listed in the figure legend.

[LINE 111] The more appropriate term is ‘sex’ in reindeer. Gender refers to identity.

[LINE 139] How did you decide to separate trot speeds into 2 categories? Were speeds bimodal? Was it arbitrary?

[LINE 141] I suggest defining what a, b, c, d, and e represent or at least reference Figure 2A here

[LINE 162] In the sentence beginning ‘Relationship between GRF and time’ should begin as ‘The relationship between GRF and time’

[LINE 164] Please include vendor information for Origin

[LINE 166] This is the first time that I see that you have referred to the ‘toe joint’. Please define it here using terms you used previously or use terminology from Figure 2.

[LINE 173] “a positive joint moment supports the weight in the normal standing state” needs to be rephrased to be an accurate statement (and I think you want to say ‘body weight’ instead of ‘the weight). I think you’re just trying to suggest that in a lot of species, a positive joint moment as an extensor moment, but that is not the case for reindeer at every joint. I would also suggest that while Vereecke et al 2008 [36] is a fine paper, I’m not sure it’s the most appropriate for citing generalities about extensor moments. That said, I think it would be reasonable to just delete the sentence on line 173 and just state which joints have positive moments for extension and for flexion.

[LINE 205] The sentence beginning “The data of speed” should probably be “Speed data” or “Animal velocity”. Additionally, “normalized by the Froude number and the nondimensionalized speed u” should probably be “normalized by Froude number” since Froude number is a nondimensionalized speed. How did you measure speed? Did you use a joint marker? Was this a mean value from TD to LO?

[LINE 234] Wouldn’t you expect these variables to differ in a walk vs a trot and at different speeds? Is
this different than what you would predict based on other ungulates?

[LINE 312] “Reindeer have permanently enhanced MCP joints” – Relative to what?

[LINE 319] Do you present data on contact time to substantiate this?

[LINE 320] ‘…positive and negative works’ should be ‘positive and negative work’

[LINE 322] Does improved elastic energy storage mean increased?

[LINE 332] Is the hypothesis that increased ROM in reindeer toes is an ecological adaptation a result of the findings of this paper or an existing hypothesis?

[LINE 352] “…for stability and propulsion through other forms”. Can you elaborate on what you mean by forms?

[LINE 388-389] If the authors are going to say that these results underlie bionic foot design of extreme environment robots, I think you will need to elaborate on this. Or remove this statement.

[LINE 360, 366, 367] There are some unnecessary placements of “the” – ‘the foot’ ‘the horse knees’

[LINE 362] I’m not sure that you need to include a discussion of human foot mechanics. I think it would be fine to just discuss mechanics in horses. But, if the authors feel it is important, then they can keep it in.

---

## Round 0.2 · Minor Revisions

The 2 reviewers have provided positive reviews of the MS and now mainly just proofreading changes are needed. PeerJ does not provide those services such as copyediting as a standard service and we cannot accept the MS until it is at a suitable standard of proofreading. Please conduct another round of revision to ensure it is to a good standard. But the science is deemed sound so we look forward to being able to accept this paper.

Reviewer 1 ·

Basic reporting

There remain a few wording issues throughout the manuscript.

For example:

Line 99. I would change the question to something like "Does the work done by or on the forelimb change with speed?"

Line 319. A suggestion would be to change this line to "Locomotor strategies change depending on the speed and/or gait adopted by the animal." Trying to avoid using 'different' twice in the sentence. This appears a couple of other times in the manuscript. It is a small issue, but relates to the readability of the doc.

I would have still liked to see the same axis scale (maximum, minimum and intervals) used in Figures 6,7, and 8. This would allow for a more visual comparison of the presented data in addition to the interpretation as to the values.

Experimental design

Improved and sufficient.

Validity of the findings

Explanation is sufficient.

Additional comments

Thank you to the authors for the extensive revisions to the document. I appreciate the time and effort that are associated with doing these revisions.

Reviewer 2 ·

Basic reporting

As before, in general, the grammar and writing are acceptable. There are a few instances where corrections are advisable, primarily in sentences added to this revision. The authors have met criteria for references, background, professional article structure, figures and tables. They have also provided the raw data. The results are self-contained and relevant to the introduction in the article. In response to reviewer comments, the authors have added questions to frame what they did in their study. I would have preferred specific hypotheses, but these are acceptable.

Experimental design

This is original primary research that is within the aims and scope of the journal. The research questions are relevant and meaningful. The authors state how this study fills a gap in the literature. The authors provided a rigorous investigation of reindeer forelimb mechanics using inverse dynamics and appear to have met institutional ethical requirements. The methods have been described sufficiently to allow for replication and improvements were made in their description in response to reviewer comments.

Validity of the findings

The findings in this study are valid, robust, and are sound. The authors have indicated what statistical methods they used (F-test), but did not offer a lot of detail. The authors conclusions are now connected to specific research questions posed in the introduction.

Additional comments

Overall, I feel that the authors have improved their manuscript with the suggested changes from the reviewers. However, there are a number of grammatical issues, primarily with the sentences that were added in this revision which will require some proofreading. I have made a few comments by referencing line numbers in the manuscript that I added for easier reference. I will upload this file.

Abstract

[Line 22] Unnecessary ‘the’. Should read ‘increased with rising locomotor speed’

[Line 24] Missing ‘the’. Should read ‘…net joint power related to the vertical GRF’

Introduction

[Line 46] Trunk shouldn’t be plural. Should read ‘…in their elastic feet to drive the trunk forward…’

There are still some grammatical and proofreading issues with this manuscript. I have pointed out the first few but many are in sentences that have been added to this revision.

[Line 167] This is likely a translation issue, but I think it would be more appropriate to say that the females were ‘excluded’ rather than ‘abandoned’.

[Line 447] The ‘Wrist joints as a stabilizer and a pusher’ section didn’t need to be completely removed, I just suggested removing references to human locomotion since I didn’t feel it was relevant.

Annotated reviews are not available for download in order to protect the identity of reviewers who chose to remain anonymous.

---

## Round 0.3 · Minor Revisions

The MS is much improved, especially readability. I recommend moving the "Forecast" text into the Conclusions. The statistical analysis in Methods could still use a little more detail; referring the reviewer to a prior paper is not enough. The full dataset from the statistical analyses should be provided with the paper to maximize the openness of the study for repeatability. Otherwise, the paper seems ready.

---

## Round 0.4 · accepted · Accept

I recommend a little final polishing of the English (e.g. last sentence of Conclusions is awkward) but the MS is acceptable now -- congratulations!